# Feasibility and Preliminary Response of a Novel Training Program on Mobility Parameters in Adolescents with Movement Disorders

**DOI:** 10.3390/healthcare13243251

**Published:** 2025-12-11

**Authors:** Phuong T. M. Quach, Gordon Fisher, Byron Lai, Christopher M. Modlesky, Christopher P. Hurt, Collin D. Bowersock, Ali Boolani, Harshvardhan Singh

**Affiliations:** 1Department of Physical Therapy, University of Alabama at Birmingham, Birmingham, AL 35294, USA; phuong@uab.edu (P.T.M.Q.); cphurt@uab.edu (C.P.H.); 2Department of Human Studies, University of Alabama at Birmingham, Birmingham, AL 35294, USA; grdnfs@uab.edu; 3Department of Pediatrics, University of Alabama at Birmingham, Birmingham, AL 35294, USA; byronlai@uab.edu; 4Department of Kinesiology, University of Georgia, Athens, GA 30602, USA; christopher.modlesky@uga.edu; 5Human Performance and Nutrition Research Institute, Oklahoma State University, Stillwater, OK 74077, USA; collin.bowersock@okstate.edu (C.D.B.); ali.boolani@okstate.edu (A.B.)

**Keywords:** cerebral palsy, spina bifida, balance, intervention, exercise, neuromotor disorder

## Abstract

**Background**: There is a critical need for feasible, non-equipment based, safe, and cost-effective exercise interventions to promote muscle strength, dynamic postural balance, and independent mobility in adolescents with cerebral palsy (CP) or spina bifida (SB). **Objectives**: This study aimed to examine the feasibility and preliminary response of a novel exercise program: Functionally Loaded High-Intensity Circuit Training (FUNHIT) and conventional High-Intensity Circuit Training (HIT) in adolescents with CP/SB. **Methods**: Enrolled participants were allocated to FUNHIT or HIT or Controls in our randomized control trial. The interventions were delivered 2×/week × 4 weeks. Feasibility was assessed through process, operational, and scientific metrics. Outcome measures included maximum walking speed, Four Square Step Test (FSST), Timed Up and Go (TUG) and its dual-task variants, Lateral Step-Up Test (LSUT), Fear of Falling (FoF) and physical activity (PA) questionnaires. **Results**: We tested 5 participants (1 CP, 4 SB) in our study. Recruitment and retention rates were acceptable (63% enrollment, 100% retention and adherence). FUNHIT (n = 2) participants showed improvements in maximum walking speed (8–12%), FSST (15–29%), LSUT (22–33%), and TUG (4%). The HIT participant (n = 1) demonstrated improved TUG dual-task performance (40%) and FSST (30%) only. Control participants (n = 2) had varied changes (from 0–24%) in mobility, strength, balance. No adverse events were reported. Participants successfully followed (100%) the prescribed exercise dosage over the four-week period. **Conclusions**: FUNHIT and HIT are feasible and safe interventions for adolescents with ambulatory CP and SB who retain motor function, showing promising preliminary improvements in muscle strength, dynamic balance, and independent mobility. Our findings need to be validated in larger samples.

## 1. Introduction

Neuromotor impairments in pediatric populations present significantly challenge independent mobility and dynamic balance. Among the most common causes of childhood motor impairment are cerebral palsy (CP) and spina bifida (SB), both of which can result in significant challenges with gait and mobility, though they arise from different neurological origins. CP is a term describing a group of permanent disorders of movement and gait, caused by a non-progressive disruptions in the developing fetus or infant brain, which can lead to upper motor neuron syndrome [1]. On the other hand, spina bifida (SB) is a neural tube defect that causes a lesion in the lower motor neuron, flaccid paralysis or paresis, sensory loss below the lesion, and frequently reduced or absent reflexes depending on lesion level [2,3]. Despite these distinct neurological origins, both CP and SB often converge on similar functional outcomes, including muscle weakness, impaired balance, and reduced mobility [1,3].

The varying degrees of damage to the brain associated with CP and spinal circuitry associated with SB result in multiple significant impairments in motor function, walking abilities, and dynamic balance [4,5,6]. For example, children affected with CP and SB often exhibit declined postural control mechanisms, reduced stride length and gait speed, and increased dependence on assistive devices for mobility [7,8], which explains their decreased independent physical activity with increasing age [5]. This is important because reduced independent physical activity often leads to secondary health issues, including musculoskeletal deformities and increased risk of other chronic diseases such as osteoporosis, hypertension, and obesity in those populations [9,10,11].

Given these substantial functional/pathological limitations, it is essential to explore effective and feasible interventions targeting these populations early in life. While numerous studies have addressed interventions for children with CP/SB during early childhood, fewer have specifically focused on adolescents [12,13,14,15]. This is critical because transition into adolescence and then adulthood further exacerbates the already present neuromotor difficulties in children with CP and SB [5,16]. Alarmingly, almost 25% of independent ambulatory adolescents with CP or SB lose their independent ambulation status around the age of 19–21 years [5]. We know that adolescence represents a crucial transitional phase from childhood to adulthood, characterized by significant physical, cognitive, and psychological changes/challenges distinct from earlier and later life stages [17]. During this time, the muscles, bones, and nervous system undergo rapid changes, potentially altering the neuromuscular control strategies established in childhood [18]. Consequently, interventions aimed at improving dynamic postural balance and muscular strength during adolescence can potentially help mitigate future functional decline and support a more seamless progression into adulthood.

There is a lot of evidence showing the effect of various interventions aimed at enhancing muscle strength, balance, and mobility in children and adolescents with CP and SB [19,20,21,22,23]. However, most of the reported effective programs depend on costly, high-end technology-based equipment such as motorized treadmills, robotic gait trainers, or stationary resistance machines, which can pose accessibility, usability, and safety issues [19,20,21,22,23,24,25]. Thus, the limited community transferability of such training programs may be an issue. Furthermore, these accessibility issues could be specifically problematic for families residing in rural areas. We also know that families with children with CP or SB have additional caregiver burdens [26,27,28]. Thus, there is a critical need for affordable, low-technology, safe, feasible, independently driven and non-machine-based exercise interventions that can deliver improvements in muscle strength, dynamic balance, and enhance independent mobility in populations with CP and SB.

There are some non-machine-based interventions that can improve postural balance and muscle strength in children and adolescents with CP or SB [29,30,31,32]. Among these interventions, High-Intensity Circuit Training (HIT) is a well-established and widely used exercise program designed to improve cardiovascular fitness, strength, and endurance [30,32]. It involves performing exercises in short bursts at maximum effort, with minimal rest periods in between [32]. This method is effective for enhancing overall cardiovascular fitness and metabolic function [30,32]. However, HIT does not specifically address the need for improved neuromotor control which is crucial for promoting independent physical activity in individuals with neuromotor impairments. These individuals often require targeted interventions to improve not just endurance, but also foundational muscle strength and the neural pathways that govern movement. Progressive resistance training (PRT) offers an evidence-based method to achieve these outcomes. Through the gradual increase in mechanical loads on the musculoskeletal system, PRT has been demonstrated to promote significant improvements in muscle strength, motor unit recruitment, and intermuscular coordination among individuals with neuromotor disorders [33,34,35]. To combine these benefits, we have developed Functionally Loaded High-Intensity Circuit Training (FUNHIT), a novel training approach that integrates the principles of HIT PRT without the use of any machine. Unlike conventional HIT, FUNHIT uses various functional movements taken from daily/leisure activities performed in the traditional HIT pattern and with progressively increasing resistance achieved through weighted vests. FUNHIT specifically emphasizes maintaining control of the center of mass over a moving base of support, which is particularly relevant for improving dynamic balance and independent mobility. Notably, FUNHIT is adaptable to various settings, highly accessible, and sustainable while enhancing the neuromotor control and postural dynamic balance needed for enhancing independent movement. What distinguishes FUNHIT from other interventions is its integration of three core training principles: (1) task-oriented functional movements; (2) progressive resistance through the use of weighted vests; and (3) adherence to a high-intensity circuit training (HIT) structure. In contrast, other interventions have typically employed only progressive resistance training (PRT) or functional movement-based protocols, such as the Loaded Sit-to-Stand, which lack the motor pattern challenge inherent in HIT [36,37]. Thus, the primary aim of this study was to evaluate the feasibility and preliminary response of our new exercise protocol (FUNHIT) in adolescents with mixed motor impairments, specifically those with functional ambulatory CP or SB. Since we used the same movement patterns of FUNHIT with HIT, we also sought to determine the feasibility and preliminary response of HIT in adolescents with mixed motor impairments.

## 2. Materials and Methods

### 2.1. Study Design and Participant Recruitment

Our study utilized a randomized controlled trial with parallel group allocation suitable for assessing feasibility in early-stage research. Participants were recruited through convenience sampling. Recruitment involved advertisements placed on clinical trial websites, flyers posted around the University of Alabama at Birmingham (UAB) campus buildings, referrals from healthcare providers, and word-of-mouth recommendations. Interested individuals reached out to research staff via email or phone and were screened for eligibility through telephone interviews. A single non-blinded tester used a simple randomization technique via a computer-generated coding to randomly allocate the enrolled participants to different arms in our study before their baseline testing. All the testing, including the exercise intervention sessions, was performed in a university laboratory setting.

Inclusion criteria included: (1) adolescents aged 11–21 years, (2) confirmed diagnosis of spastic cerebral palsy (CP) or spina bifida (SB), (3) ability to walk independently (for participants with CP: an ambulatory status classified as Gross Motor Function Classification System [GMFCS] levels I–II; for participants with SB: ambulation with incomplete lesions or lumbosacral-level lesions), (4) capacity to understand and follow verbal instructions in English, and (5) willingness of participants and their parents to commit to the full duration of the study. Exclusion criteria were: (1) having undergone orthopedic surgery or botulinum toxin A injections in the lower limbs within the past 6 months, (2) participation in structured lower limb exercise training within the past 6 months, (3) plans to engage in structured exercise training outside the study, (4) presence of cardiac, pulmonary, visual, auditory, or other physical conditions that could hinder study participation, and (5) any additional medical conditions that could interfere with performance of independent physical activity.

As this study focused on feasibility rather than efficacy, no formal sample size calculation was performed. Our study was reviewed and approved by the UAB Institutional Review Board ensuring compliance with ethical standards for research involving human participants. We registered our study on ClinicalTrials.gov (Trial registration number: NCT05865418, date of registration: 18 May 2023).

### 2.2. Participant Characteristics

A total of five participants (one CP and four SB), ages ranging from ~14 to ~20 years, were included in the study as shown in Table 1.

### 2.3. Exercise Dose Determination

Two baseline sessions were performed for exercise dose determination, which was customized for each type of exercise movement and each participant. Thus, we minimized the learning effect. At the 1st baseline, all participants were asked to perform 3 sets (30 s × 3) of each of the 5 exercises (Figure 1A) with the instruction: do as many as you can with the correct form in 30 s. Notably, all the exercises were informed by functional movements and included movements typically used by children for their daily and leisurely activities. Chair squats were used to strengthen lower-limb muscles essential for sit-to-stand transfers, which are often impaired in children with cerebral palsy [38]. Split squats were included to improve hip extensor muscles, challenge dynamic balance, and improve motor control for asymmetrical tasks such as stair climbing [39]. Heel raise was included to strengthen the calf muscles, helping to create a stronger “push-off” with each step, addressing common deficits in propulsive force generation [40]. Side-steps were included to strengthen the hip abductors, which are key muscles often weakened in children with CP and strongly associated with impaired dynamic balance, gait performance, and gross motor function [41]. Countermovement jump was included to enhance explosive power, rate of force development, and the stretch-shortening cycle, which have been shown to improve strength and balance in youth with CP [42]. Together, these exercises aimed to enhance balance, strength, and mobility through movements reflecting real-life functional demands. Before any exercise dose determination testing, a single tester demonstrated all the exercises to our participants. The tester showed both the correct and incorrect ways of doing the exercises. Then, all the participants tried the exercise movement 1–5 times to familiarize themselves. Rest periods of 30s and 90s were provided between each set and exercise-type, respectively. After the end of each set, we recorded the participant’s rating of perceived exertion (RPE).

For the participants who successfully completed all sets of exercises, we recorded the repetitions per set for each exercise and used the lowest number of repetitions as their starting dose for that specific exercise. At their 2nd baseline session, which was within 7 days of their 1st baseline visit, we asked them to perform the pre-determined exercise dose per exercise. A successful completion with the pre-determined exercise dose per exercise at the 2nd baseline visit confirmed their unique starting dose. In cases where the participant indicated that they could achieve higher repetition for 3 sets for any specific exercise, we performed another assessment to adjust the exercise dosage per the new number achieved.

For the participants who failed to complete any of the three sets for the full 30 s at the 1st baseline, we recorded the maximum number of repetitions that were performed. Then, we reduced that number by 10% and tested the participant’s performance again for the same exercise. These participants also performed the pre-determined exercise dose per exercise at their 2nd baseline visit for their unique exercise dose determination. Thus, we were able to determine customized dose for each participant per exercise.

### 2.4. Exercise Intervention

Three participants (1 CP and 2 SB) completed a 4-week intervention (2×/week) involving either HIT (n = 1, SB) or FUNHIT (n = 2, CP + SB). All three participants performed the same exercises: split squats, chair squats, side steps, heel raise, and countermovement jumps (Figure 1A) for the same duration. Each participant’s exercise dose and progression for each session followed a personalized order based on their baseline exercise dose per exercise. Each participant in the intervention groups (HIT or FUNHIT) was instructed to increase their repetitions for each exercise by two at the start of every week, beginning on day 1 week 1 of the intervention. A metronome was used to help them maintain a consistent pace. Participants were encouraged to complete more repetitions in the last set of each exercise if they felt comfortable. The exercise was progressed only for those participants who successfully completed the determined exercise volume for the specific week. Exercise progression was initiated on Day 2 of each week by increasing the number of repetitions by 2. This process was repeated until the end of the intervention period.

During the exercise dose determination/baseline sessions, we asked each participant to rate the 5 exercises based on their perceived difficulty. We used this information to design a personalized exercise sequence for each participant with the least difficult exercise as the starting exercise. Next, we arranged the remaining 4 exercises to alternate between higher and lower perceived difficulty levels to ensure that the ‘difficult’ and ‘easy’ exercises followed each other. This was mainly to ensure that the participants could finish the full exercise session without undue fatigue interfering with their exercise performance. The description of each exercise is outlined in Appendix A.

All the exercises were the same between the HIT and FUNHIT group, with the FUNHIT group performing these exercises while wearing weighted vests (Fun and Function, LLC; made in USA, see Figure 1B). The FUNHIT participants started their 1st exercise session at 2.5% of their body weight and increased by 2.5% weekly, reaching 10% by week 4. Each session lasted approximately 20 min. All the exercise sessions were delivered in the same university laboratory setting, where we conducted the pre-and post-assessments, under the direct one-on-one supervision of a trained research assistant. The control group (n = 2, SB) was instructed to continue their usual care, which consists of their routine daily activities and regular diets, and they participated only in pre- and post-intervention testing.

### 2.5. Feasibility Metrics

Feasibility metrics are shown in Table 2.

### 2.6. Assessments

The following assessments were used to examine the preliminary response of our interventions (FUNHIT and HIT) in our study:The Four Square Step Test (FSST) utilized four canes arranged in a cross on the floor, forming four squares, where participants were instructed to step forward, sideways, and backward in a specific sequence as quickly as possible without touching the canes, with the time recorded in seconds using a stopwatch to assess dynamic balance and coordination [43].The Lateral Step-Up Test (LSUT) required participants to stand beside an adjustable step (height set at low level (~17 cm) and high level (~37 cm)) and step laterally onto it with one leg, fully extending the knee, then return to the starting position, repeating this for 20 s, with the number of completed steps counted to measure lower limb strength [44].The Timed Up and Go Test (TUG) involved participants rising from a standard chair, walking 3 m to a marked cone, turning 180 degrees, and returning to sit, with time recorded in seconds using a stopwatch to evaluate functional mobility [45].The TUG with counting backward required participants to count aloud by subtracting 3 from a random number between 20–80 while performing the standard TUG [46].The TUG with a full cup of water required participants to carry a full plastic cup of water (around 250 mL) while performing the standard TUG and not spilling the water [47].The Unrestrained Maximum Walking Speed Test employed a 6-m-long pressure-sensitive mat (ProtoKinetics Zeno™ Walkway Gait Analysis System; ProtoKinetics LLC, Havertown, PA, USA), where participants started two steps before and walked past the mat at their fastest safe speed, with the mat capturing spatiotemporal gait measures (such as speed (m/s), step length (cm), and cadence (steps/min)) to measure their functional mobility [48].The Restrained Maximum Walking Speed Test, on the same Zeno™ walkway system, required participants to start by standing on the mat and then walking at their fastest safe speed and stopping at a marked position on the mat near its endpoint [48].The PROMIS Physical Activity (PA) Questionnaire consists of 10 questions assessing the participant’s physical activity over the past 7 days. Responses are scored on a scale from 1 to 5 (1 = no days, 2 = 1 day, 3 = 2–3 days, 4 = 4–5 days, 5 = 6–7 days), with higher total scores indicating greater levels of physical activity during the previous week [49].The Pediatric Fear-of-Falling (FoF) Questionnaire includes 34 questions evaluating the participant’s fear of falling during various daily activities. Responses are rated on a scale from 0 to 2 (0 = never, 1 = sometimes, 2 = always) or marked as N/A (not applicable), with higher total scores indicating greater fear of falling [50].

All the tests were assessed by the same consistent tester. All physical assessments were tested at two baseline timepoints to minimize learning effects, whereas the self-reported questionnaires were administered only at Baseline 2.

Statistical Analyses:Due to the pilot and feasibility design of our study, we report our results of feasibility and treatment response metrics as absolute numbers. Mean and SD (Standard deviation) are reported for exercise dosage determination and progression. Treatment responses for the pre-post changes are reported as percentages.

## 3. Results

### 3.1. Feasibility Study

#### 3.1.1. Process Metrics

Twelve children were contacted about the study during our recruitment period (August 2023–August 2024), and, after screening, eight of them (66.7%) were eligible to participate. Of these eight eligible children, five (63%) agreed to enroll in the study. Three children (37%) declined participation due to various reasons: one was involved in summer sports or school activities, and two had parents who were unavailable to provide transportation to the lab. Among the five enrolled participants, two heard about the study through a clinical trial website, and three enrolled through word of mouth.

All five enrolled participants (100%) completed both pre-intervention assessments (Baseline 1 and Baseline 2) as well as the post-intervention assessments. Among these participants, all three children in the intervention group successfully completed 100% of the intervention protocol, including all prescribed sets and cycles (2 sessions/week for 4 weeks), indicating excellent adherence. The description is presented in Appendix B: CONSORT Flow Chart.

#### 3.1.2. Operational Feasibility

The median duration from the initial participant contact to enrollment was approximately 7 days. Eligibility screening took an average of 20 min per participant. The pre-intervention assessment phase averaged approximately 2.75 h, with the intervention group requiring roughly 3 h and the control group about 2.5 h. The post-intervention assessment phase lasted approximately 2.25 h for both groups, as the intervention group did not require dose determination during this phase.

Intervention sessions averaged 24 min (13 min of exercises HIT/FUNHIT and 11 min of rest period) each, conducted twice weekly over 4 weeks, resulting in a total intervention time of 192 min. Participants spent an average of 15 and 6 min completing the FoF and PA questionnaires, respectively. Additionally, personnel spent approximately five hours downloading Zeno walkway data, with subsequent processing of these data requiring approximately 28 h.

The total compensation for the participants in the intervention groups was USD 430 and for the control group was USD 210. We had a team of one doctoral level student, one faculty, and three undergraduate students to help with this study regarding personnel cost. Since we already had a well-established research laboratory, the extra material costs involved buying weighted vests (~USD 150), snacks and beverages for our participants.

#### 3.1.3. Scientific Evidence

Throughout the study, there were no adverse events, serious adverse events, or clinical emergencies reported. All participants in the intervention groups (100%) showed outstanding adherence, with all three successfully completing the prescribed frequency (two sessions/week), duration (four weeks), and all sets and cycles as planned.

Furthermore, intervention participants demonstrated preliminary improvements in exercise performance, reflected by their successful achievement of the prescribed exercise dose from baseline to post-intervention (Table 3), with average preliminary improvements ranging from approximately 18.6% for the side steps to 69.1% for the chair squats (Table 4). Intervention responses were evaluated through delta and percentage changes in mobility, muscle strength, dynamic balance, FoF, and PA outcomes. All participants in the intervention group exhibited positive responses, showing preliminary improvements as % changes shown in Table 5, Table 6, Table 7, Table 8 and Table 9.

### 3.2. Exercise Repetitions

Participants demonstrated positive responses across the four-week training program, with preliminary improvements varying by exercise and individual (Table 4). For the chair squats, increases were 60.00% (P01), 60.98% (P02), and 86.21% (P03), with an average improvement of 69.06%. For the split squats, performance improved by 39.02% (P01), 43.18% (P02), and 46.67% (P03), averaging 42.96%. For the side steps, improvements were more modest, with increases of 14.29% (P01), 6.19% (P02), and 35.29% (P03), with an average of 18.59%. For the jump, improvements were 62.86% (P01), 13.41% (P02), and 42.42% (P03), averaging 39.57%. Finally, for the heel raise, performance increased by 18.18% (P01), 85.71% (P02), and 24.00% (P03), with an average improvement of 42.63%.

### 3.3. Preliminary Data

In terms of walking speed measures (Table 5), unrestrained maximum walking speed preliminary improved in all intervention participants, with P01 and P02 (FUNHIT group) showing improvements of 8% and 12%, respectively. In contrast, P03 (HIT group) exhibited only a small improvement (2%). For the restrained maximum walking speed test, participant P01 showed a 30% improvement, while P02 declined by 12%. Participant P03 also showed a 14% improvement in walking speed performance, whereas the control participants (P04 and P05) showed varied changes (10% and −3% respectively) in their restrained maximum walking speed.

Preliminary improvement in the performance of the LSUT (lateral step-up test) at both low (0.017 m) and high (0.037 m) step heights was noted in the FUNHIT group (Table 6). P01 and P02 demonstrated test performance improvements ranging from 27% to 33% at low height; and from 22% to 30% at high height at the end of four weeks. Conversely, P03 (HIT group) showed no improvement. Control group participants showed varied ranges of improvements (4–24%) at the end of four weeks.

As shown in Table 7 and Table 8, for the timed functional tests (FSST, TUG, TUG with counting, and TUG with a cup), negative deltas and percentages indicated preliminary improvements, with reduced times demonstrating enhanced performance. Our preliminary data show that the FSST improved for the intervention participants, notably, P03 (30%), P01 (29%), and P02 (15%) at the end of four weeks. The TUG test showed mild improvements (4–11%), most notably for P03 with 11% at the end of four weeks. For the cognitive dual-task test (TUG counting), improvements occurred for P03 (40%) and P02 (35%), but P01’s performance declined by 15% at the end of four weeks. The TUG cup test also showed improvements in the intervention participants, notably P03 (17%) and P01 (15%), and smaller improvement for P02 (2%) at the end of four weeks.

Regarding questionnaires (Table 9), fear of falling (FoF) scores varied, with preliminary improvement in confidence (lower scores) noted for P01 (42%), while P02 (56%) and P03 (20%) demonstrated increased fear scores. Control participants showed minimal changes in FoF questionnaire (−11% and 0%). Physical activity (PA) scores showed mixed responses, with positive responses for P02 (41%) and modest improvements for the control group, but slight reductions for P01 (9%) and P03 (23%). Similar with FoF questionnaire, control participants also showed minimal changes in PA questionnaire (16% and 6%).

## 4. Discussion

### 4.1. Feasibility

Our study showed that both FUNHIT and HIT exhibited preliminary promise with improving walk speed, muscle strength, and dynamic balance in adolescents with CP and SB. Notably, although the neurological origins of Cerebral Palsy (UMN lesion) and Spina Bifida (LMN lesion) are distinct [1,2,3], the SB participants in our pilot study were functioning ambulators with lumbosacral-level lesions. This criterion ensured they retained voluntary neuromuscular activation (paresis, not plegia) in the muscle groups targeted by the training, making them physiologically capable of responding to the strengthening and balance intervention. However, FUNHIT and HIT may differ in the degree of their responses. Our study recruitment process revealed a moderate eligibility rate, with approximately two-thirds of screened children meeting the eligibility criteria. Excellent feasibility in terms of attendance, exercise progression, and exercise response of FUNHIT and HIT in adolescents with CP and SB was another main finding of our study.

Notably, among eligible participants, the enrollment rate was 63%. However, several challenges emerged. While the clinical trial website and flyers were initially utilized, they appeared to be less effective than anticipated. In contrast, word-of-mouth recruitment improved the study enrollment, suggesting that personal recommendations from trusted sources such as healthcare providers or friends play a crucial role in participant recruitment. Indeed, previous literature shows that personal referral often generates higher trust and comfort among potential participants, especially for clinical or exercise intervention studies [51].

A specific recruitment challenge was noted for the CP population, with notably limited enrollment, potentially due to multiple concurrent studies recruiting from the same population pool from the same geographical area. Additionally, the requirement for onsite intervention sessions may have proven to be restrictive, as logistical factors such as travel distance and scheduling conflicts (summer camps and school sessions) affected participation. In the future, online interventions might mitigate these barriers, allowing for more flexibility in scheduling, particularly for families with busy or seasonal routines. However, the adherence rate with online/tele rehabilitation is a cause for concern, with a recent study reporting 43% adherence at week 4 in a four-week intervention program in adolescents with CP [52]. Indeed, the adherence-related limitations of home-based rehabilitation strategies in children with CP are well known in the literature [53,54]. Thus, we posit that using novel, user-friendly, cost-effective, customizable, cosmetically appealing wearable technology in combination with home-based rehabilitation program could provide an exciting strategy to improve adherence addressing neuromotor impairments in adolescents with CP and SB.

We postulate that, given the challenges of time commitment with onsite participation, seamless integration of researchers with the adolescent’s school system/teachers may provide a highly novel environment for greater participation in research studies. This is in line with previous evidence showing that involving teachers and leveraging school system for exercise training program in children can enhance the effectiveness of the training program [55]. Moreover, a systematic and meta-analysis review also suggested that school-based physical activity can substantially enhance cognitive outcomes, especially executive functioning [56]. A school-based exercise intervention may also have good feasibility for exercise adherence as shown by 75% exercise adherence in a previous study utilizing group-based intervention [57]. Furthermore, presenting the potential benefits of study participation such as increased independent mobility for adolescents with cerebral palsy, reduced caregiver burden, and improved workplace productivity to the employers of the adolescent’s family may help enhance recruitment efforts. Finally, monetary compensation to cover the cost of travel and leave from work may further incentivize and attract more participants [58]. Enhanced compensation could offset travel and scheduling inconveniences, making the study more appealing and feasible for potential participants.

Notably, no adverse events or clinical emergencies occurred during the study, demonstrating the safety and excellent tolerability of our intervention study. All intervention group participants demonstrated excellent adherence to the twice-weekly intervention protocol and completed all prescribed sets (100%) and repetitions (100%), confirming that the proposed frequency and intensity were feasible and acceptable for this population. The intervention sessions were short, lasting about 24 min each session, which likely explained high participant adherence and lessened their participation burden. Moreover, our intervention produced meaningful preliminary improvements, with average exercise repetition increases ranging from approximately 18.59% for Side steps exercises to 69.06% for Chair squats.

The FUNHIT participants wore weighted vests, with the weight increasing by 2.5% of their body weight each week. Despite this added challenge, P01 and P02 showed strong consistency and preliminary improvement, especially in exercises such as chair squats and jumps, where they continued to improve their repetitions until the very last exercise session. This suggests that progressively increasing external resistance through weighted vests did not negatively affect repetition adherence and may have even positively influenced performance by systematically building muscle strength and neuromotor control [59]. Similarly, P03, in the HIT group without a weighted vest, continued to improve their repetitions for each exercise until the last exercise session. Thus, we infer that meaningful improvements can also be achieved without external loads, especially if the exercise program is individually dose-determined and customized per perceived difficulty of exercises. Setting incremental weekly targets provided clear motivation for participants, likely prompting them to discover personal strategies to surpass their previous limits, activate their muscles more efficiently, or utilize their skills optimally during exercises. Thus, the intervention may have generated self-efficacy in our population [60]. However, we did not assess any self-efficacy measures in our study.

Finally, Lai et al. (2020) identified three research priorities that serve as recommendations for interventions among children and adolescents with disabilities: (1) achieving long-term outcomes, (2) developing precision-based interventions, and (3) establishing scalable strategies [61]. For the first priority (long-term outcomes), we understand that our four-week follow-up is short but position this feasibility study as the essential first step required to justify a future, longer-term trial. For the second priority (developing a precision-based intervention), we frame our FUNHIT program as a move toward a more “precision-based” approach, as it is specifically designed to target each participant’s capability through the dose determination and their body weight. For the third priority (establishing scalable strategies), we now emphasize that a key strength of our study is its direct contribution to this priority. By testing a low-cost, non-machine-based intervention, we are exploring a model that is inherently more scalable and transferable to community or school settings compared to other programs.

### 4.2. Preliminary Findings

Our preliminary data suggest improvements in unrestrained and restrained walk speed, dynamic balance, and mobility, further supporting the feasibility and potential positive response of the interventions (FUNHIT and HIT) in adolescents with CP and SB. Greater improvements in unrestrained maximal walk speed in FUNHIT vs. HIT might be explained by the additional increases in muscle strength and neuromotor control due to incremental weighted vests. The added resistance could promote muscular adaptations, translating into greater functional capacity for speed and movement control [62]. Thus, combining the principle of progressive resistance training and high intensity interval training might offer greater clinical benefits for gait in individuals with neurological impairments. Interestingly, the restrained maximum walking speed test results revealed varied responses. Participant P01’s improvement (30%) demonstrates that the FUNHIT intervention could effectively facilitate a participant’s ability to control and stabilize after rapid movements. In contrast, participant P02’s reduced performance (12% decline) might suggest fatigue, motivational factors, or a need for additional instructional cues to effectively improve deceleration control.

In step count tests (LSUT), both FUNHIT participants (P01 and P02) showed preliminary improvements, potentially due to the progressive resistance and its positive impact on lower limb strength, stability, and endurance. The absence of change for the HIT participant (P03) suggests that incremental loading might be particularly beneficial for enhancing neuromuscular properties.

Timed functional assessments (FSST; TUG and its dual-task variations) revealed meaningful differences across participants. The FSST showed notable improvements in P03 (30%) and P01 (29%), with P02 also demonstrating a moderate gain (15%). These findings are consistent with earlier observations regarding the positive impact of high-intensity and progressive resistance training on dynamic balance and neuromuscular function [63]. Improvements in the standard TUG test were modest (ranging from 4% to 11%), aligning with the FSST results and reflecting gradual mobility improvements. Outcomes for the dual-task versions of TUG varied, particularly within the FUNHIT group. In the TUG counting test, P03 and P02 exhibited substantial improvements of 40% and 35%, respectively, suggesting enhanced cognitive–motor integration. However, P01 experienced a 15% decline, which may reflect cognitive fatigue or reduced ability to manage the combined cognitive and physical demands introduced by the weighted vest.

The TUG-dual tasking response variability within the FUNHIT group suggests that, while progressive loading can enhance physical performance, it may also contribute to fatigue or limit performance during cognitively demanding tasks. Similarly, the TUG with a cup test showed improvements in P03 (17%) and P01 (15%), indicating better postural control and coordination during functional tasks involving object manipulation. In contrast, P02 demonstrated only a minimal improvement (2%), further supporting the possibility that cumulative fatigue from resistance loading may affect performance consistency in some participants. Interestingly, the substantial reductions in FSST and TUG counting times for P03, who participated in the HIT group without external loading, highlight the potential for high-intensity, non-weighted training to facilitate neuromuscular adaptations and improve cognitive-motor integration [30,32]. In comparison, the varied responses in the FUNHIT group may reflect an interaction between the added physical demands of resistance training and the participants’ cognitive capacity, which can influence outcomes in dual-task settings.

The FoF score demonstrated highly interesting data. P01 showed substantially reduced fear of falling (42%), possibly explained by enhanced walk speed, muscle strength, and dynamic balance. However, P02 and P03 reported increased FoF, by 56% and 20%, respectively, despite improvements in walk speed, muscle strength, and dynamic balance. This suggests that psychological adaptations may not be in sync with physical improvements in adolescents with CP and SB. Thus, we posit that improvements in muscle strength and dynamic balance as noted in our study may paradoxically increase FoF. Control participants showed minimal change, suggesting relative stability in their perceived FoF without intervention [64]. PA questionnaire results also varied across participants. P02 demonstrated the greatest improvement, with a 41% increase in reported activity, likely reflecting translation of improved walk speed, dynamic balance, and mobility. Control participants showed modest increases (6–16%), possibly due to natural variation or seasonal effects. In contrast, P01 and P03 reported decreases in PA levels (9% and 23%, respectively), which may suggest fatigue, limited opportunity for activity during the reporting period, or temporary disengagement following the intervention. A recent review showed that children who participate in exercise intervention studies could compensate by doing less PA at home during the intervention period [65]. Moreover, these results should be interpreted with caution, as the questionnaire captures activity levels only within the previous seven days and may not reflect broader behavioral trends.

### 4.3. Limitations

This study has several limitations that should be considered. First, the sample size was small, with only five participants in total. Such a limited number of participants restricts the generalizability of our findings and limit the statistical power to attribute changes to intervention. Nevertheless, the study was designed as a feasibility investigation, and the results may provide valuable insights into the feasibility of tests that can capture our treatment response on muscle strength, dynamic balance, and mobility in our population.

Second, we tested both CP and SB in our study despite them being different neurological conditions. In fact, four of the five participants had SB, limiting our ability to explore pathology-based feasibility responses to ourintervention. However, shared moblity-related challenges [5,7,8] made testing adolescents with CP and SB relevant for our pilot testing of the intervention.

Third, baseline differences among participants were not controlled or examined due to the feasibility nature of this study. Variables such as severity of condition, muscle stiffness, or baseline physical function were not matched across groups, which may have influenced post-intervention outcomes. For instance, participants in the control group appeared to demonstrate stronger baseline function. This, combined with the fact that we could not formally monitor or quantify their ‘usual care’ activities, potentially affected comparative results and obscured the true effects of the intervention. Without assessing and accounting for these baseline disparities, it is difficult to draw clear conclusions about the effectiveness of the FUNHIT and HIT protocols. Fourth, the study had some methodological limitations such as lack of follow-up data and accelereomter-estimated physical activity data.

Overall, while the findings from this feasibility study are promising and suggest the potential benefits of the interventions, larger and more balanced studies with well-matched groups are needed to confirm efficacy and generalize results to broader populations. More importantly, this study offers valuable, practical strategies for the successful implementation of those future, larger-scale trials. For instance, our findings on recruitment clearly demonstrated that provider referrals were significantly more effective than passive methods like flyers or websites. This provides a clear directive for future studies to prioritize building strong partnerships with hospitals, schools, and local clinics to ensure a successful recruitment pipeline.

Furthermore, our findings on adherence and safety highlighted both notable strengths and key challenges. The high tolerability of the one-on-one supervised format fostered a safe and motivating environment, leading to excellent retention among participants who enrolled. However, the need for participants to travel to the laboratory emerged as a major barrier to recruitment, with many families reporting transportation and scheduling difficulties. These observations suggest that a hybrid model combining in-lab sessions with telehealth delivery may offer an optimal solution. In such a model, participants could complete initial assessments and training in the lab, followed by home-based practice supported through remote monitoring, thereby enhancing both recruitment and adherence. Moreover, the use of a weighted vest proved to be an affordable, portable, and effective tool, making it particularly suitable for integration into this hybrid approach for home or school settings.

## 5. Conclusions

In summary, this feasibility study supports the viability and potential positive preliminary response of both FUNHIT and HIT interventions. Moderate recruitment success combined with excellent retention and fidelity of exercise dosing progression, with no adverse events, demonstrates the viability of the next stage of research with our study. These interventions, especially when incorporating incremental resistance, show promise in improving functional performance, dynamic balance, and muscle strength, with further research needed to refine intervention protocols, optimize individual adaptations, and address cognitive–motor integration.

## Figures and Tables

**Figure 1 healthcare-13-03251-f001:**
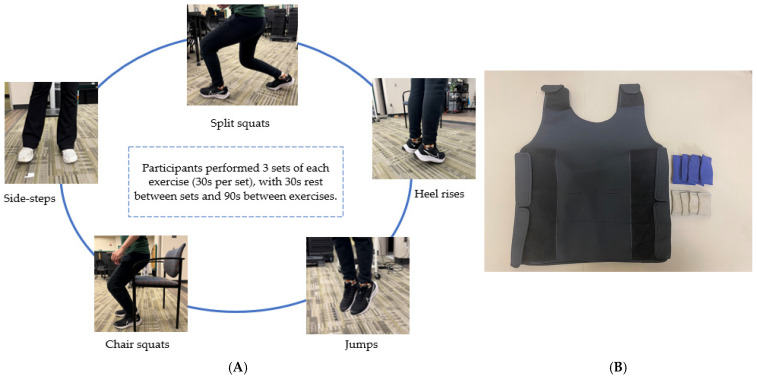
(**A**) Representative images of the exercise movements that were performed by participants in our study. A detailed description of these exercises is provided in Appendix A. (**B**) Weighted vest used in FUNHIT intervention.

**Table 1 healthcare-13-03251-t001:** Participant characteristic.

ID	Age (y)	Diagnosis	Group	Sex
P01	20.2	CP	FUNHIT	Male
P02	14.1	SB	FUNHIT	Female
P03	14.8	SB	HIT	Male
P04	15.6	SB	Control	Male
P05	14.2	SB	Control	Male

*CP, cerebral palsy; SB, Spina bifida; FUNHIT, Functionally Loaded High-Intensity Circuit Training; HIT, High-Intensity Circuit Training; P01–05 represent our 5 participants.*

**Table 2 healthcare-13-03251-t002:** Feasibility metrics and assessment approaches.

Metric	Outcome	Assessment Approach
**Process metrics:** evaluate participant recruitment, retention, and adherence	1. Participant screening and enrollment metrics 2. Participant completion rate 3. Compliance rate	1. Number of individuals screened, found eligible, and successfully enrolled. 2. Refusal rate and documented reasons for non-participation. 3. Completion rate for baseline/pre-intervention assessments. 4. Retention rate through the 4-week intervention period. 5. Overall study completion rate, based on the post-intervention assessment.
**Operational metrics:** evaluate the time and financial resources required for the study, as well as the data management and safety reporting procedures	1. Time spent on recruitment, pre/post assessment, and intervention phases 2. Time and effort for participant data collection and personnel data processing. 3. Financial resources necessary to conduct the study	1. Duration of participant recruitment (from initial contact to enrollment) (days) 2. Duration of eligibility screening (minutes) 3. Time spent on the pre-assessment phases (hours) 4. Time spent on the post-assessment phase (hours) 5. Time spent on the intervention phase (hours) 6. Time for participants to complete FoF Questionnaire (minutes) 7. Time for participants to complete PA Questionnaire (minutes) 8. Personnel time to download and process Zeno walkway path data (hours) 9. Total cost (USD)
**Clinical metrics:** evaluate safety, burden, and treatment effect	1. Adverse events, serious adverse events, and clinical emergencies 2. Participants’ ability to adhere to the prescribed intervention dosage and repetitions throughout the study 3. Intervention effects measured via: FSST TUG, TUG with a cup, TUG with counting LSUT at low and high level Zeno Walkway FoF Questionnaire PA Questionnaire	1. Number of adverse events, serious adverse events, and clinical emergencies 2. Assessment of participants’ adherence to the prescribed intervention frequency, duration, and repetitions. 3. Intervention responses were assessed and changes in dynamic balance and muscle strength outcome variables

*LSUT, Lateral Step-Up Test; FoF, Fear of Falling; PA, Physical activity; FSST, Four square step test; TUG, Timed Up and Go.*

**Table 3 healthcare-13-03251-t003:** Exercise repetitions from pre-intervention to week 4 intervention.

Exercise	ID	Pre-Intervention		Week 1		Week 2		Week 3		Week 4	
		Baseline 1 (Mean ± SD)	Baseline 2 (Mean ± SD)	Day 1 (Mean ± SD)	Day 2 (Mean ± SD)	Day 1 (Mean ± SD)	Day 2 (Mean ± SD)	Day 1 (Mean ± SD)	Day 2 (Mean ± SD)	Day 1 (Mean ± SD)	Day 2 (Mean ± SD)
Chair squats	P01	16.67 ± 0.58	16.00 ± 0	17.00 ± 1	18.00 ± 0	20.00 ± 1.73	20.67 ± 0.58	22.00 ± 0	22.00 ± 0	23.33 ± 1.15	26.67 ± 0.58
	P02	13.67 ± 0.58	14.00 ± 0	16.00 ± 0	16.00 ± 0	17.33 ± 1.15	18.00 ± 0	19.33 ± 1.15	20.00 ± 0	21.33 ± 1.15	22.00 ± 0
	P03	9.67 ± 1.53	11.33 ± 1.15	12.00 ± 0	12.00 ± 0	13.33 ± 1.15	14.00 ± 0	15.33 ± 1.15	16.00 ± 0	17.33 ± 1.15	18.00 ± 0
Split squats	P01	13.67 ± 1.53	14.00 ± 0	15.67 ± 0.58	15.00 ± 1	15.00 ± 0	15.33 ± 0.58	16.33 ± 1.15	17.00 ± 0	18.33 ± 1.15	19.00 ± 0
	P02	14.67 ± 2.31	15.00 ± 1.73	14.33 ± 1.15	15.00 ± 0	16.33 ± 1.15	17.00 ± 0	18.33 ± 1.15	19.00 ± 0	20.33 ± 1.15	21.00 ± 0
	P03	15.00 ± 4.58	16.00 ± 0	16.00 ± 0	16.00 ± 0	17.33 ± 1.15	18.00 ± 0	19.33 ± 1.15	20.00 ± 0	21.33 ± 1.15	22.00 ± 0
Side steps	P01	35.00 ± 5.57	36.00 ± 0	38.00 ± 0	37.67 ± 0.58	39.33 ± 1.15	36.67 ± 1.15	37.33 ± 0.58	38.67 ± 1.15	38.67 ± 1.15	40.00 ± 3.46
	P02	37.67 ± 2.31	34.33 ± 2.08	34.00 ± 0	34.00 ± 0	35.33 ± 1.15	36.00 ± 0	37.33 ± 1.15	38.00 ± 0	39.33 ± 1.15	40.00 ± 0
	P03	17.00 ± 2.65	16.67 ± 1.53	17.67 ± 1.15	17.00 ± 0	18.33 ± 1.15	19.00 ± 0	20.33 ± 1.15	21.00 ± 0	22.33 ± 1.15	23.00 ± 0
Jumps	P01	11.67 ± 0.58	11.00 ± 0	12.33 ± 0.58	13.00 ± 0	13.00 ± 0	15.00 ± 0	15.00 ± 0	17.00 ± 0	15.67 ± 1.15	19.00 ± 0
	P02	27.33 ± 3.21	22.67 ± 0.58	25.00 ± 0	25.00 ± 0	26.33 ± 1.15	27.00 ± 0	28.33 ± 1.15	29.00 ± 0	30.33 ± 1.15	31.00 ± 0
	P03	22.00 ± 5.20	25.00 ± 0	25.00 ± 0	25.00 ± 0	26.33 ± 1.15	27.00 ± 0	28.33 ± 1.15	29.00 ± 0	30.33 ± 1.15	31.33 ± 0.58
Heel raise	P01	22.00 ± 1	21.00 ± 0	22.00 ± 1	22.00 ± 1	22.33 ± 1.15	22.67 ± 0.58	23.67 ± 1.15	25.00 ± 0	25.00 ± 2	26.00 ± 1.73
	P02	18.67 ± 2.08	19.00 ± 2.65	19.33 ± 1.15	29.67 ± 0.58	31.33 ± 1.15	30.67 ± 1.53	33.33 ± 1.15	34.00 ± 0	35.00 ± 1.73	34.67 ± 1.15
	P03	25.00 ± 2.65	23.67 ± 1.53	24.00 ± 0	24.00 ± 0	25.33 ± 1.15	26.00 ± 0	27.33 ± 1.15	28.00 ± 0	29.33 ± 1.15	31.00 ± 0

*SD, Standard deviation.*

**Table 4 healthcare-13-03251-t004:** Exercise repetitions from baseline to week 4 intervention.

Exercise	ID	% Improvement from Baseline 1 to Week 4 Day 2
Chair squats	P01	60.00%
	P02	60.98%
	P03	86.21%
	**Average: 69.06%**	
Split squats	P01	39.02%
	P02	43.18%
	P03	46.67%
	**Average: 42.96%**	
Side steps	P01	14.29%
	P02	6.19%
	P03	35.29%
	**Average: 18.59%**	
Jumps	P01	62.86%
	P02	13.41%
	P03	42.42%
	**Average: 39.57%**	
Heel raise	P01	18.18%
	P02	85.71%
	P03	24.00%
	**Average: 42.63%**	

**Table 5 healthcare-13-03251-t005:** Results of unrestrained and restrained maximum walking speed tests.

ID	Group	Unrestrained Maximum Walking Speed (cm/s)		Delta	Percentage Change	Restrained Maximum Walking Speed (cm/s)		Delta	Percentage Change
		Baseline 2	Posttest			Baseline 2	Posttest		
P01	FunHIT	130.029	140.53	10.501	**8%**	57.952	75.19	17.238	**30%**
P02	FunHIT	120.602	135.522	14.92	**12%**	75.294	65.884	−9.41	**−12%**
P03	HIT	71.397	72.957	1.56	**2%**	41.2	47.121	5.921	**14%**
P04	Control	209.874	210.985	1.111	**1%**	92.292	101.862	9.57	**10%**
P05	Control	217.511	239.697	22.186	**10%**	86.453	83.469	−2.984	**−3%**

**Table 6 healthcare-13-03251-t006:** Results of LSUT at low level and high level.

ID	Group	LSUT at Low Level (Step Counts)		Delta	Percentage Change	LSUT at High Level (Step Counts)		Delta	Percentage Change
		Baseline 2	Posttest			Baseline 2	Posttest		
P01	FunHIT	11	14	3	**27%**	10	13	3	**30%**
P02	FunHIT	9	12	3	**33%**	9	11	2	**22%**
P03	HIT	7	7	0	**0%**	7	7	0	**0%**
P04	Control	22	26	4	**18%**	23	24	1	**4%**
P05	Control	21	26	5	**24%**	20	24	4	**20%**

*LSUT, Lateral Step Up Test.*

**Table 7 healthcare-13-03251-t007:** Results of FSST and TUG.

ID	Group	FSST (Seconds)		Delta	Percentage Change	TUG (Seconds)		Delta	Percentage Change
		Baseline 2	Posttest			Baseline 2	Posttest		
P01	FunHIT	11.14	7.89	−3.25	**−29%**	8.92	8.6	−0.32	**−4%**
P02	FunHIT	8.7	7.39	−1.31	**−15%**	10.23	9.85	−0.38	**−4%**
P03	HIT	29.73	20.73	−9	**−30%**	14.64	12.96	−1.68	**−11%**
P04	Control	5.56	5.67	0.11	**2%**	9.9	8.92	−0.98	**−10%**
P05	Control	6.23	5.55	−0.68	**−11%**	8.44	8.25	−0.19	**−2%**

*FSST, Four square step test; TUG, Timed Up and Go.*

**Table 8 healthcare-13-03251-t008:** Results of TUG with counting and TUG with holding a cup.

ID	Group	TUG with Counting (Seconds)		Delta	Percentage Change	TUG with Holding a Cup (Seconds)		Delta	Percentage Change
		Baseline 2	Posttest			Baseline 2	Posttest		
P01	FunHIT	13.32	15.29	1.97	**15%**	18.66	15.87	−2.79	**−15%**
P02	FunHIT	15.84	10.33	−5.51	**−35%**	15.42	15.09	−0.33	**−2%**
P03	HIT	24.73	14.93	−9.8	**−40%**	22.07	18.23	−3.84	**−17%**
P04	Control	10.02	10.03	0.01	**0%**	12.55	11.87	−0.68	**−5%**
P05	Control	11.68	9.38	−2.3	**−20%**	10.33	10.09	−0.24	**−2%**

*TUG, Timed Up and Go.*

**Table 9 healthcare-13-03251-t009:** Results of FoF questionnaire and PA questionnaire.

**ID**	**Group**	**FoF Questionnaire** **(Score)**		**Delta**	**Percentage Change**	**PA Questionnaire** **(Score)**		**Delta**	**Percentage Change**
		**Baseline 2**	**Posttest**			**Baseline 2**	**Posttest**		
P01	FunHIT	19	11	−8	**−42%**	33	30	−3	**−9%**
P02	FunHIT	9	14	5	**56%**	22	31	9	**41%**
P03	HIT	10	12	2	**20%**	35	27	−8	**−23%**
P04	Control	9	8	−1	**−11%**	32	37	5	**16%**
P05	Control	22	22	0	**0%**	34	36	2	**6%**

*FoF, Fear of Falling; PA, Physical activity.*

## Data Availability

The data presented in this study are available on request from the corresponding author. The data are not publicly available due to privacy concerns.

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
