# Peer review of "Feasibility and Preliminary Response of a Novel Training Program on Mobility Parameters in Adolescents with Movement Disorders"

_healthcare, 2025, doi:10.3390/healthcare13243251_

Round 1

Reviewer 1 Report

Comments and Suggestions for Authors
  • Strengths: You list small sample size, imbalance in diagnoses, and uncontrolled baseline differences.
  • Missing limitations:
    • Lack of assessor blinding.
    • Short follow-up (only 4 weeks).
    • Recruitment bias: mostly SB participants, not representative.
    • Use of subjective questionnaires (FoF, PA) without triangulation from objective activity measures (e.g., accelerometers).
Comments on the Quality of English Language
  • Grammatical mistakes
    •  small errors (“There is lot of evidence…”; “University of Alabama at Brimingham” instead of “Birmingham”).
    • Inconsistent tense use—switches between past and present.
    • “enrollment reached approximately 63%” → should be “enrolment rate was 63%.”
  • Formatting:
    • Journal style not consistently followed (e.g., author affiliations layout, “Received/Revised/Accepted” placeholders left unfilled).
    • Figures (1A and 1B) are referenced but not visible in the provided text—ensure proper image placement and captions.
  • References:
    • Good coverage but uneven formatting (e.g., some have full DOI links, others truncated).
    • Check reference style consistency according to Healthcare journal guidelines.

Author Response

We are sincerely thankful to the reviewers and editorial team for their rigorous and insightful evaluation of our manuscript. Addressing your constructive critiques have enhanced the quality of our work. We have addressed each comment as follows with the hope that our manuscript is ready for a hopefully positive decision on its acceptance. Thank you!

For Reviewer 1:

We are sincerely thankful to the reviewers and editorial team for their rigorous and insightful evaluation of our manuscript. Addressing your constructive critiques have enhanced the quality of our work. We have addressed each comment as follows with the hope that our manuscript is ready for a hopefully positive decision on its acceptance. Thank you!

English Editing: The author team of this manuscript contains native speakers of English. In addition, the corresponding author of the submitted manuscript has >40 peer-reviewed published manuscripts in English language scientific journals. We have proofread our work and confident that our revised manuscript will be found suitable for its English language usage by the reviewers. Thank you for your time.

Comment about missing limitations:

  • Lack of assessor blinding.
  • Short follow-up (only 4 weeks).
  • Recruitment bias: mostly SB participants, not representative.
  • Use of subjective questionnaires (FoF, PA) without triangulation from objective activity measures (e.g., accelerometers).

Response: Thank you for pointing this out. I've added these missing points to the limitations section.

Comments on the Quality of English Language: about grammatical mistakes, formatting (Figures (1A and 1B) are referenced but not visible in the provided text—ensure proper image placement and captions) , fix references

Response: Thank you for your comment.

  • I have proofread the full manuscript for any language-related mistakes.
  • Figures (1A and 1B): I have checked their inclusion in the text.
  • I have changed and edited some references to make it consistent per the journal’s guidelines.

English Editing: The author team of this manuscript contains native speakers of English. In addition, the corresponding author of the submitted manuscript has >40 peer-reviewed published manuscripts in English language scientific journals. We have proofread our work and confident that our revised manuscript will be found suitable for its English language usage by the reviewers. Thank you for your time.

Reviewer 2 Report

Comments and Suggestions for Authors

Dear Authors,

The development of rehabilitation programs and development of therapeutic interventions for patients with motor disorders remains an area of ongoing debate among specialists and a challenge for all therapists.

I would like to share some personal comments, which in no way diminish the value of your study.

Questions / Suggestions:

  • Why did you choose participants with different pathologies? Cerebral palsy and spina bifida are quite different conditions (one involves UMN lesions, the other affects LMN). Wouldn’t it be better for your study to use a more homogeneous sample?
  • Since your study is about feasibility, why do you also attempt to draw outcome results from such a small group of patients? In my opinion, the focus should be only on feasibility, ideally with a larger number of participants, as you also mention (line 125).
  • As you know, feasibility studies are usually guided by specific criteria that help test both effectiveness and efficacy. Even though rehabilitation lacks clear guidelines, there are several references describing such criteria. Instead, most of your article focuses on presenting the intervention program and drawing preliminary conclusions, rather than evaluating feasibility. At this stage, I believe the main focus should be on testing feasibility, by collecting data from a larger sample.
  • You don’t mention feasibility results from similar intervention programs, which would give us useful comparison points.

In conclusion, I think your study should concentrate solely on testing the feasibility of your intervention itself, rather than drawing early conclusions about its outcomes compared to another exercise program (HIT). This is something your future research will address. Therefore, I feel that any conclusions drawn from these preliminary data are somewhat risky. You also note that in limitations.

It has been a pleasure to read your manuscript.

Author Response

We are sincerely thankful to the reviewers and editorial team for their rigorous and insightful evaluation of our manuscript. Addressing your constructive critiques have enhanced the quality of our work. We have addressed each comment as follows with the hope that our manuscript is ready for a hopefully positive decision on its acceptance. Thank you!

Comment 1: Why did you choose participants with different pathologies? Cerebral palsy and spina bifida are quite different conditions (one involves UMN lesions, the other affects LMN). Wouldn’t it be better for your study to use a more homogeneous sample?

Response 1: Thank you for your question. We agree that cerebral palsy (CP) and spina bifida (SB) represent different neurological conditions; however, our pilot study was designed to explore the feasibility and preliminary response of our exercise intervention across a spectrum of pediatric motor impairments. We had already mentioned that in the introduction in our original version. We have added new lines under limitations in the revised version addressing the same.

Comment 2: Since your study is about feasibility, why do you also attempt to draw outcome results from such a small group of patients? In my opinion, the focus should be only on feasibility, ideally with a larger number of participants, as you also mention (line 125).

Response 2: Thank you for this valuable feedback. Our intention was not to evaluate efficacy but to focus on preliminary responses to show feasibility of i) our intervention to bring a change, and ii) feasibility of tests to capture those preliminary intervention responses. Throughout the manuscript, we have highlighted the pilot nature and preliminary response of the study. We have removed the word ‘effects’, ‘outcomes’, in context of the study intervention throughout the manuscript, and added ‘preliminary’ with ‘improvements’ throughout the manuscript.

Comment 3: As you know, feasibility studies are usually guided by specific criteria that help test both effectiveness and efficacy. Even though rehabilitation lacks clear guidelines, there are several references describing such criteria. Instead, most of your article focuses on presenting the intervention program and drawing preliminary conclusions, rather than evaluating feasibility. At this stage, I believe the main focus should be on testing feasibility, by collecting data from a larger sample.

Response 3: Please see our response to the above comment. Specifically, the last paragraph of the ‘Discussion’ section was added to address your concerns.

Comment 4: You don’t mention feasibility results from similar intervention programs, which would give us useful comparison points.

Response 4: Thank you for pointing out this omission. To enhance the context of our findings, we’ve revised and expanded the Discussion section. Interestingly, we could find no information on feasibility metrics for study which were done in a similar way to us: meaning 1) on-site, 2) high intensity training with added vests, and 3) adolescents with CP and spina bifida. Thus, we are unable to provide data comparing the feasibility results from other studies for a valid comparison. However, we found an online/tele rehabilitation study reporting their adherence rate, and we compared their findings with our results in the discussion.

Comment 5: In conclusion, I think your study should concentrate solely on testing the feasibility of your intervention itself, rather than drawing early conclusions about its outcomes compared to another exercise program (HIT). This is something your future research will address. Therefore, I feel that any conclusions drawn from these preliminary data are somewhat risky. You also note that in limitations.

Response 5: Please see our response above (response 3).

Reviewer 3 Report

Comments and Suggestions for Authors

Dear Editor of the Journal

Hello

The title reflects the text well

In the abstract, it is better to mention the actual results in the results section

In the introduction, it is better to mention the necessity of the work more

In the method section, how was the sampling method?

In the equipment used, what country was it made?

It is better to use the profit and loss chart

In the discussion, you should pay more attention to the reasons

References should be updated

Author Response

We are sincerely thankful to the reviewers and editorial team for their rigorous and insightful evaluation of our manuscript. Addressing your constructive critiques have enhanced the quality of our work. We have addressed each comment as follows with the hope that our manuscript is ready for a hopefully positive decision on its acceptance. Thank you!

Comment 1: In the abstract, it is better to mention the actual results in the results section.

Response 1: Thank you for your suggestion. We are mindful of the word limit of the abstract so we have shown main results in the results section of the abstract.

Comment 2: In the introduction, it is better to mention the necessity of the work more.

Response 2: We agree. As you suggested, we have expanded the Introduction to more strongly emphasize the necessity and importance of this research.

Comment 3: In the method section, how was the sampling method?

Response 3: We have clarified this in the " Study design and participant recruitment" subsection of the Materials and Methods part. The manuscript now states that we used a convenience sampling method, which is a common and practical approach for an initial feasibility study of this nature.

Comment 4: In the equipment used, what country was it made?

Response 4: We have added this detail to the "Exercise Intervention" subsection of the Materials and Methods part. The text now indicated that the equipment was made in the USA.

Comment 5: It is better to use the profit and loss chart.

Response 5: We appreciate the reviewer’s suggestion. However, we believe there may be a misunderstanding, as a “profit and loss chart” is a financial tool typically used in business to illustrate revenue and expenses. Such a chart is not relevant to our clinical feasibility study, which does not involve financial data or outcomes. If the reviewer intended to recommend a visual representation of the intervention’s “pros and cons” or the balance between benefits and burdens for participants, we believe our detailed discussion of feasibility outcomes (e.g., high adherence, absence of adverse events) already captures these aspects of the intervention’s value and acceptability. Therefore, we have not included this type of chart.

Comment 6: In the discussion, you should pay more attention to the reasons.

Response 6: Thank you for this valuable advice. We have substantially revised the Discussion section to provide a deeper analysis of our findings.

Comment 7: References should be updated.

Response 7: We’ve reviewed and updated our reference list.

Reviewer 4 Report

Comments and Suggestions for Authors

General comments

This study examined the feasibility and preliminary response of an exercise program on mobility parameters in adolescents with movement disorders. The study prescribed two interventions functionally loaded high-intensity circuit training, and conventional high-intensity circuit training over a 4-week period, with 5 participants recruited to the study. Improvements are observed for several outcomes measures, but the authors acknowledge that findings need to be validated in a larger sample. Young people with disability are rarely involved in physical activity research, so the authors should be commended for conducting this important work. While the manuscript is generally well supported by relevant literature, I have provided some further recommendations. Please note that any papers recommended in the report are for reference purposes only and are not mandatory. You are welcome to cite and reference other relevant papers related to this topic.

A previous review of physical activity interventions involving children and youth with disability identified several methodological limitations, including issues with generalizability, transferability, and scientific rigor. The review also outlined three key research priorities for future work: i) achieving long-term and sustainable post-intervention outcomes, ii) developing precision-based interventions, and iii) establishing scalable intervention and recruitment strategies. Can the authors please clarify how the current study addresses these previously identified methodological limitations or research priorities? It would strengthen the manuscript to explicitly integrate this discussion throughout the manuscript, citing the work by Lai and colleagues.

Reference:

  • Lai, B., Lee, E., Wagatsuma, M., Frey, G., Stanish, H., Jung, T., & Rimmer, J. H. (2020). Research trends and recommendations for physical activity interventions among children and youth with disabilities: a review of reviews. Adapted Physical Activity Quarterly, 37(2), 211-234. https://doi.org/10.1123/apaq.2019-0081

Specific comments

Introduction

The introduction is well written and provides a compelling case for effective, low-cost treatment targeting this population. The authors suggest that high-intensity circuit training/high-intensity interval training (HIT) is a feasible approach, and has showed efficacy. While the authors acknowledge these programs have traditionally involved the use of machines (allowing for more controlled movements), functionally loaded HIT (without machines) has been suggested as an alternative. The sentence “It involves performing exercises on a machine in short bursts at maximum effort, with minimal rest periods in between” (lines 83-84) is slightly confusing and appears to disrupt the flow of argument. For example, the previous two sentences in the same paragraph highlight that there are some non-machine based interventions, however this definition mention the use of machines.

Further, previous interventions involving children and adolescents living with a range of disability have incorporated similar non-based machine methods, that would address neuromotor control through the application of functional exercises that engage multiple muscle groups and require coordination (e.g., body weight squats). I’d recommended that the authors clearly describe how their intervention differs from those previously delivered.

Methods

General: Did the authors involve formal progression criteria to decide whether to proceed with further testing? The CONSORT guidelines for feasibility trials specify including progression criteria to determine whether or not to progress with the next stage of research. “We expect that when pilot or feasibility studies are proposed by applicants, or specified in commissioning briefs, a clear route of progression criteria to the substantive study will be described”.

https://www.equator-network.org/reporting-guidelines/consort-2010-statement-extension-to-randomised-pilot-and-feasibility-trials/

General: Did the research team collect data regarding participants’ acceptance/satisfaction of the program?

General: The use of two baseline testing periods is a thoughtful strategy to limit the learning effect, particularly for physical measures. However, given the small sample size (and lack of statistical testing), firm conclusions cannot be drawn regarding participants’ baseline values. In addition, while this approach is understandable for physical measures where a learning effect may influence results, it is less clear why it was also applied to questionnaire responses. If this was not the case for questionnaire responses, can the authors please clarify this in the methods as results later indicate (baseline 2).

2.4 Exercise intervention: It would be museful if more detail could be provided for the intervention including where sessions were delivered and any supervision required for sessions.

Line 199: Can the authors provide additional information regarding usual care procedures for those in the control groups?

Table 2: The resource and management metrics provide useful information about the assessment approaches. However, the categorisation is somewhat unclear. For example, time spent on pre- and post-assessment phases is listed under resource metrics, whereas the time for participants to complete FoF and PA questionnaires are included under management metrics. Since both sets of measures reflect assessment procedures, it would be clearer to group them together under a consistent heading.

Table 2: The term ‘Scientific metrics’ is confusing in the context of this study, particularly as no statistical analyses were performed due to the small-scale study. The authors also include descriptive details for adverse events, serious adverse events, and clinical emergencies under this heading. I suggest the authors revise this terminology so that it is more appropriate.

2.6 Assessments: Citations are not provided for the participant mobility outcomes. Can the authors please provide these to justify why the specific measures were used.

Results

General: The results are generally well presented, however, the manuscript repeatedly highlights “notable performance gains”. While these observations are interesting, they should be interpreted with extreme caution due to the very small sample size (n=5) and lack of statistical analyses.

Lines 283-285: It’s useful that the authors have provided the toral intervention time. Is there any information to support how long usual care treatment occurred for those in the control group during the study period?

Line 286: Please include the total cost, as mentioned in Table 2.

Discussion

General: The discussion does read slightly too favourably and begins with ‘excellent feasibility’. While the results of the study are interesting, the small sample size cannot be discounted and therefore results may not be generalisable. I’d recommended that the authors provide a general summary paragraph highlighting the main study findings prior to addressing specific subsections in detail.

Paragraph 3: The authors highlight the importance of the school setting for increasing engagement and participation in research studies. This is a valid suggestion as schools can provide greater reach for paediatric population, and may mitigate barriers to participation of community-based interventions. I believe this paragraph could be strengthened by providing reference to school-based programs involving young people with disability, as there is currently a lack of integrated relevant research in this paragraph. While some citations are provided, these are not specific to young people living with disability. Please see a recent review for relevant school-based physical activity research:

  • Leahy, A., A., Robinson, K., Eather, N., Smith, J., J., Hillman, C., H., Beacroft, S., Mazzoli, E., & Lubans, D., R. (In press). School physical activity interventions for children and adolescents with disability: Systematic review and meta-analysis of effects on academic, cognitive, and mental health outcomes. Journal of Physical Activity and Health. https://doi.org/10.1123/jpah.2025-0052

4.3 Limitations: Lines 514-515: The authors highlight that their study results may provide valuable insight into potential effectiveness and implementation of intervention strategies. Can the authors please elaborate or provide further detail here regarding the specific strategies.

Author Response

We are sincerely thankful to the reviewers and editorial team for their rigorous and insightful evaluation of our manuscript. Addressing your constructive critiques have enhanced the quality of our work. We have addressed each comment as follows with the hope that our manuscript is ready for a hopefully positive decision on its acceptance. Thank you!

General Comments

Comment: Can the authors please clarify how the current study addresses these previously identified methodological limitations or research priorities [from Lai et al., 2020: i) achieving long-term outcomes, ii) developing precision-based interventions, and iii) establishing scalable strategies]? It would strengthen the manuscript to explicitly integrate this discussion...

Response: Thank you for this excellent recommendation and for bringing the pivotal review by Lai et al. (2020) to our attention. We agree that framing our study within these research priorities adds significant value. We have now integrated this discussion into Discussion section.

Specific Comments

Introduction part

Comment: The sentence “It involves performing exercises on a machine in short bursts at maximum effort, with minimal rest periods in between' (lines 83-84) is slightly confusing and appears to disrupt the flow of argument...”

Response: Thank you for pointing out this lack of mistake. We have removed the word “on a machine”.

Comment: "I’d recommended that the authors clearly describe how their intervention differs from those previously delivered [e.g., body weight squats]."

Response: Thank you for your recommendation. We have now added a more explicit description in the ‘Introduction’ section to differentiate our intervention. We clarify that its novelty lies in the specific combination of a structured HIT format with progressive resistance training and task-oriented functional movements that differ from interventions focusing on single PRT.

Methods part

Comment (General): " Did the authors involve formal progression criteria to decide whether to proceed with further testing? The CONSORT guidelines for feasibility trials specify including progression criteria to determine whether or not to progress with the next stage of research. We expect that when pilot or feasibility studies are proposed by applicants, or specified in commissioning briefs, a clear route of progression criteria to the substantive study will be described”.

Response: We’ve followed the CONSORT guidelines and with our results we think it can be proceeded with further testing.

Comment (General): "Did the research team collect data regarding participants’ acceptance/satisfaction of the program?"

Response: No, we did not document the acceptance/satisfaction of the program. In our future studies, we plan to do that. Thank you for your question.

Comment (General): " The use of two baseline testing periods is a thoughtful strategy to limit the learning effect, particularly for physical measures. However, given the small sample size (and lack of statistical testing), firm conclusions cannot be drawn regarding participants’ baseline values. In addition, while this approach is understandable for physical measures where a learning effect may influence results, it is less clear why it was also applied to questionnaire responses. If this was not the case for questionnaire responses, can the authors please clarify this in the methods as results later indicate (baseline 2).”

Response: The two-baseline approach was applied primarily to the physical measures to account for learning effects. This two-baseline approach was not applied to the self-report questionnaires. The questionnaires were administered only once, at the second baseline session (Baseline 2). We have updated the Methods section to make this distinction clear.

Comment (2.4 Exercise intervention): It would be useful if more detail could be provided for the intervention including where sessions were delivered and any supervision required for sessions.

Response: We’ve expanded this section to include these details. The text now specifies that all the exercise sessions were delivered in the same university laboratory setting, where we conducted the pre-and post-assessments, under the direct one-on-one supervision of a trained research assistant.

Comment (Line 199): "Can the authors provide additional information regarding usual care procedures for those in the control groups?"

Response: We have added more details to the Methods. Participants in the control group were instructed to continue their 'usual care,' which consisted of their routine daily activities and regular diets.

Comment (Table 2): “Table 2: The resource and management metrics provide useful information about the assessment approaches. However, the categorization is somewhat unclear. For example, time spent on pre- and post-assessment phases is listed under resource metrics, whereas the time for participants to complete FoF and PA questionnaires are included under management metrics. Since both sets of measures reflect assessment procedures, it would be clearer to group them together under a consistent heading.”

Response: Thank you for your suggestion. We have now merged resource and management metrics into “Operational metrics”.

Comment (Table 2): "The term “Scientific metrics” is confusing... I suggest the authors revise this terminology so that it is more appropriate."

Response: We’ve changed the term “Scientific metrics” to the " Clinical Metrics"

Comment (2.6 Assessments): "Citations are not provided for the participant mobility outcomes. Can the authors please provide these citations "

Response: We have now added the appropriate citations for each of the mobility outcome measures.

Results part

Comment (General): " The results are generally well presented, however, the manuscript repeatedly highlights “notable performance gains”. While these observations are interesting, they should be interpreted with extreme caution due to the very small sample size (n=5) and lack of statistical analyses.

Response: We’ve replaced phrase like "notable performance gains" with more cautious and appropriate language "positive trends"; and “gains” for “improvements” or “increases”

Comment (Lines 283-285): "Is there any information to support how long usual care treatment occurred for those in the control group?"

Response: As noted in our response regarding the Methods, we did not formally track the duration of usual care. We have now explicitly stated this as a limitation in the Discussion section.

Comment (Line 286): "Please include the total cost, as mentioned in Table 2."

Response: We’ve added the total compensation for the control group and the intervention groups, and the cost of a weighted vest in the Results section.

Discussion part

Comment (General): "The discussion does read slightly too favorably and begins with ‘excellent feasibility’. While the results of the study are interesting, the small sample size cannot be discounted and therefore results may not be generalizable. I’d recommended that the authors provide a general summary paragraph highlighting the main study findings prior to addressing specific subsections in detail.”

Response: We have restructured the 1st paragraph of the discussion section.

Comment (Paragraph 3): "I believe this paragraph could be strengthened by providing reference to school-based programs involving young people with disability."

Response: Thank you for this excellent suggestion and for providing a highly relevant reference. We have rewritten this paragraph to integrate the findings to our discussion.

Comment (4.3 Limitations): Lines 514-515: The authors highlight that their study results may provide valuable insight into potential effectiveness and implementation of intervention strategies. Can the authors please elaborate or provide further detail here regarding the specific strategies.

Response: We have expanded this point in the Limitations section to be more concrete about potential effectiveness and implementation of intervention strategies.

Round 2

Reviewer 2 Report

Comments and Suggestions for Authors

Comments to Authors:

Dear Authors,

Comment No 1.

Since you explained that “… our pilot study was designed to explore the feasibility and preliminary response of our exercise intervention across a spectrum of pediatric motor impairments …”, I think this should be clearly reflected in the abstract (lines 60–61) and in the main text. Even though your preliminary study included children with Cerebral Palsy (CP) and Spina Bifida (SB), the aim should clearly state that the study focuses on motor impairments in general, and not specifically on children with CP and SB, who were included only to assess the feasibility of the new training program.

My comment regarding the inclusion of SB does not relate to it being a different pathology but to the nature of the condition, and I will explain. From the first sentence of your manuscript (lines 58–61), it appears that your novel training program aims to promote muscle strength and dynamic postural balance, among other outcomes, in adolescents with SB. However, all individuals with SB, regardless of severity, have peripheral nerve involvement, which means they cannot achieve voluntary neuromuscular activation - i.e. motor unit recruitment. This is incompatible with the training program you aim to evaluate, as it focuses on techniques for muscle strengthening and dynamic postural balance, activities to which these children cannot respond due to the nature of their disorder. They cannot respond to muscle strength training techniques, nor achieve dynamic balance, because their muscles do not activate.

In the Conclusions, you state that “FUNHIT and HIT are feasible and safe interventions for adolescents with CP and SB with promising preliminary improvements in muscle strength, dynamic balance …”. Allow me, but UNDER NO CIRCUMSTANCES can children with SB respond to such a program.

This is the main reason for my objection - not the fact that you included a different pathology, which in principle is acceptable, but the nature of this specific pathology and the fact that the training program focuses on techniques to which these patients cannot respond. I believe this issue can be resolved by not including individuals with Lower Motor Neuron (LMN) lesions.

Comment No 2.

You cannot use the GMFCS for children with SB to determine functional level. The GMFCS is designed primarily for children with cerebral palsy and is not appropriate for children with SB.

Thank you.

Author Response

Comment No 1:

Since you explained that “… our pilot study was designed to explore the feasibility and preliminary response of our exercise intervention across a spectrum of pediatric motor impairments …”, I think this should be clearly reflected in the abstract (lines 60–61) and in the main text. Even though your preliminary study included children with Cerebral Palsy (CP) and Spina Bifida (SB), the aim should clearly state that the study focuses on motor impairments in general, and not specifically on children with CP and SB, who were included only to assess the feasibility of the new training program.

My comment regarding the inclusion of SB does not relate to it being a different pathology but to the nature of the condition, and I will explain. From the first sentence of your manuscript (lines 58–61), it appears that your novel training program aims to promote muscle strength and dynamic postural balance, among other outcomes, in adolescents with SB. However, all individuals with SB, regardless of severity, have peripheral nerve involvement, which means they cannot achieve voluntary neuromuscular activation - i.e. motor unit recruitment. This is incompatible with the training program you aim to evaluate, as it focuses on techniques for muscle strengthening and dynamic postural balance, activities to which these children cannot respond due to the nature of their disorder. They cannot respond to muscle strength training techniques, nor achieve dynamic balance, because their muscles do not activate.

In the Conclusions, you state that “FUNHIT and HIT are feasible and safe interventions for adolescents with CP and SB with promising preliminary improvements in muscle strength, dynamic balance …”. Allow me, but UNDER NO CIRCUMSTANCES can children with SB respond to such a program.

This is the main reason for my objection - not the fact that you included a different pathology, which in principle is acceptable, but the nature of this specific pathology and the fact that the training program focuses on techniques to which these patients cannot respond. I believe this issue can be resolved by not including individuals with Lower Motor Neuron (LMN) lesions.

Response: Thank you very much for your insightful comments and constructive feedback. We have carefully revised the manuscript accordingly.

First, in the Introduction, we clarified that the primary aim of this pilot study was to evaluate the feasibility and preliminary response of the novel exercise protocol in adolescents with mixed motor impairments, represented by ambulatory CP and SB, rather than focusing narrowly on either diagnosis alone. This reflects our intent to target motor impairments more broadly.

Not all children with spina bifida are completely incapable of voluntary activation. In fact, there are well-established reports of improvement in gait speed, muscle strength, and walking endurance in population with spina bifida (https://pubmed.ncbi.nlm.nih.gov/29579008/ ; https://pubmed.ncbi.nlm.nih.gov/11305403/ ). We are thankful to the reviewer as this lack of clarity led to us including information on the level of our participants with spina bifida.

Thus, in the Methods section, we revised our inclusion criteria for participants with SB to specify that they were community ambulators with lumbosacral-level lesions. This criterion ensured that all included individuals retained voluntary neuromuscular activation (paresis rather than plegia) in the muscles targeted by our intervention, making them physiologically capable of responding to the training.

Third, in the Discussion, we emphasized that the SB participants in this pilot study were functioning ambulators with lumbosacral lesions who demonstrated retained voluntary muscle activation in lower limbs. This qualification addresses the concern regarding the nature of SB pathology and supports the rationale for including these individuals in this intervention.

Comment No 2.

You cannot use the GMFCS for children with SB to determine functional level. The GMFCS is designed primarily for children with cerebral palsy and is not appropriate for children with SB.

Response: Thank you for pointing this out. We have revised the inclusion criteria to specify the GMFCS requirement for participants with CP and the functional ambulatory level (incomplete or lumbosacral-level lesions) for participants with SB.

Reviewer 3 Report

Comments and Suggestions for Authors

Edited and its can be publish 

Author Response

Response: Thank you for your thoughtful comments. They were very helpful in improving the overall quality of the manuscript. We sincerely appreciate the time and effort you devoted to reviewing our paper.

Reviewer 4 Report

Comments and Suggestions for Authors

I thank the authors for addressing each of the questions raised in my review, which has improved the overall quality of the manuscript. Well done on conducting this important work.

Author Response

(The authors gave the same response as above.)
